# A pre – post quasi-experimental study of team-based learning effectiveness for Vietnamese nursing students

**Lan Duong Thi Ngoc**  [ID][☉], **Anh Nguyen Ngoc Quynh** [ID][☉]*, **Thao Hoang Thi Phuong**[☉]

University of Medicine and Pharmacy, Hue University, Hue, Vietnam

☉ These authors contributed equally to this work.
* nnqanh@huemed-univ.edu.vn

## Abstract

### Background

Team-Based Learning (TBL) is a student-centered teaching strategy designed to improve problem-solving skills, knowledge, and practical abilities. Despite its increasing use in nursing education globally, limited research has explored its effectiveness in Vietnam. This study evaluates the impact of TBL on learning outcomes, accountability, preferences, satisfaction, engagement, perceptions, and attitudes of Vietnamese nursing students during the Nursing Care for Adults with Internal Medicine course.

### Methods

A quasi-experimental pre-post study was conducted with 186 fourth-year nursing students at a nursing faculty in Vietnam during the 2023–2024 academic year. TBL was implemented in the course, and data were collected using validated instruments, including the Individual Readiness Assurance Test (i-RAT), Team Readiness Assurance Test (t-RAT), Classroom Engagement Survey (CES), and Perceived Collective Efficacy (PCE) scale. Data were collected from January to April 2024 and analyzed using SPSS version 20.0.

### Results

The mean t-RAT scores significantly exceeded i-RAT scores, increasing from 8.17 to 9.68 (t = -19.507, p < 0.001), indicating improved group performance. Students' attitudes toward teamwork showed significant improvements across all dimensions, with higher post-TBL mean scores. CES and PCE scores also increased significantly post-TBL (31.37 ± 2.002 vs. 29.54 ± 2.186; t = -8.981, p < 0.001; 4.03 ± 0.488 vs. 3.64 ± 0.461, t = -8.667, p < 0.001). Additionally, students reported positive experiences with TBL, with average scores for accountability, preference, and satisfaction at 31.19 ± 2.975, 57.10 ± 5.279, and 36.54 ± 3.815, respectively.

**Data availability statement:** All relevant data are within the paper and its Supporting Information files.

**Funding:** The author(s) received no specific funding for this work.

**Competing interests:** The authors have declared that no competing interests exist.

## Conclusions

TBL effectively enhances academic performance, teamwork attitudes, and group responsibility awareness among Vietnamese nursing students. This approach holds promise for improving nursing education in Vietnam, and educators are encouraged to expand its application to other universities and disciplines.

## Introduction

The traditional presentation method of teaching, where the lecturer is at the center of the learning process, has limitations. It limits active participation and inhibits multidimensional knowledge exchange, and lacks focus on developing students' skills in organizing and synthesizing content. To address these challenges, more active teaching methods, such as Problem-Based Learning (PBL) and Flipped Learning, have been adopted globally, including in Vietnam. These methods have proven to be highly effective. However, applying them to large groups of students while ensuring educational quality remains a challenge for educators and educational management bodies in Vietnam.

In response to these challenges, Team-Based Learning (TBL) has emerged as a promising solution. TBL organizes students into small groups, focusing on both individual and group readiness, and applying collective knowledge to solve problems [1]. It is a learner-centered approach, where instructors guide group work and assess classroom engagement to promote active learning and critical thinking [2]. TBL is widely used in medical and nursing education for its effectiveness in fostering teamwork and applying clinical knowledge. Research shows that TBL enhances test scores (i-RAT and t-RAT), improves group learning attitudes, increases confidence in teamwork, and boosts class participation. Additionally, TBL positively influences students' accountability, interest, and overall satisfaction with the learning process [2].

Given the increasing demand for nursing professionals and a significant rise in nursing student enrollment, methods like PBL are facing challenges due to limited faculty for small group teaching. TBL presents a solution for these challenges.

Despite its successful implementation worldwide, the impact of TBL on nursing students in Vietnam, particularly regarding grades, attitudes, perceptions, classroom participation, and group responsibility, remains underexplored. Positive attitudes associated with TBL include increased confidence, teamwork, idea-sharing, and its recognized role in enhancing clinical knowledge and skills, fostering effective collaboration in clinical scenarios [3]. Perceived collective efficacy, the belief that group members can achieve goals together, encourages individual engagement by boosting confidence in the group's success [4]. Classroom engagement involves focus, commitment, and active participation in learning activities [5]. Accountability in group learning refers to the responsibility members feel to contribute, complete tasks, and support the group's success [6].

Given these gaps in research, this study aims to assess the effectiveness of TBL in nursing education by using a pre- and post-study design. The focus will be on key

outcomes such as i-RAT and t-RAT scores, group learning attitudes, perceptions of group work efficacy, class participation, accountability, interest, and student satisfaction. The findings will provide valuable insights into the role of TBL in improving nursing education in Vietnam.

## Methods

### Study population

This is a quasi-experimental study with a single-group pre- and post-test design. We integrated components from the STROBE declaration to enhance the quality of reporting observational studies [7]. This study included 186 fourth-year nursing students enrolled in the Nursing Care of Adults with Internal Medicine course from June 2023 to December 2024. The study received approval from The Institutional Ethics Committee of Hue University of Medicine and Pharmacy (Approval number: H2023/110) and was conducted under the principles outlined in the Declaration of Helsinki [8]. Prior to participation, written informed consent was obtained from students.

We used G*power software to calculate the sample size to compare two means on the same group of subjects [9], with an effect factor of d = 0.25, a type I error (α) of 0.05, and a power of 0.95. The minimum sample size required was 164 students. To ensure the integrity of the study, we invited all 186 nursing students in the 2023–2024 academic year to participate in the Nursing Care for Adults with Internal Medicine course.

Among the 4th-year full-time Bachelor of Nursing students enrolled in the Faculty of Nursing, University of Medicine and Pharmacy, Hue University, all students were fully informed about the purpose of the study. All students agreed to participate, and written informed consent was obtained before their involvement. Student names were not linked to any data collected during the research, and the data were handled exclusively by the authors. The participants were students who had developed distinct facets of their professional identity and expertise. The Team-Based Learning (TBL) sessions took place between January and April 2024. Students who were absent during any part of the TBL sessions were excluded from the study. Only those enrolled in the Nursing care for adults with Internal Medicine diseases course and who agreed to participate were included. All 186 students participated in the study, with no students excluded.

### Data collection

The research process was conducted following a structured approach to implement the Team-Based Learning (TBL) method among the study participants. This process involved the following steps:

**Introduction and group division.** A meeting was organized to introduce the TBL teaching-learning method to the students and guide them through its steps. A total of 186 fourth-year nursing students from two classes, Class A (93 students) and Class B (93 students). Each class was randomly divided into 10 groups of 9–10 students, which remained unchanged throughout the TBL sessions.

**Teaching content and module design.** The subject "Nursing care for adults with internal medicine diseases" was divided into three modules: (1) Nursing care for adults with musculoskeletal diseases, (2) Nursing care for adults with digestive diseases, and (3) Nursing care for adults with kidney and urinary diseases. Three sessions, one for each module, were conducted across four teaching periods (50 minutes per period). Each module followed six core steps of the TBL method:

Preparation before class: Approximately one week before the class, students received comprehensive learning materials, including textbooks, reputable websites, and foreign-language resources from the Faculty of Nursing's library.

Individual Readiness Assurance Test (i-RAT): Students completed a 20-question multiple-choice test to assess their preparation.

Group Readiness Assurance Test (t-RAT): Groups answered the same 20 questions as i-RAT, using immediate feedback assessment techniques. The test was conducted in 15 minutes without reference materials.

Group discussions and problem clarification: Groups listed unresolved issues on a blackboard. The instructor facilitated discussions and provided explanations for unanswered questions.

Case-based problem solving: Each module concluded with situational exercises, allowing students to apply knowledge and solve case studies.

Lesson summary: The instructor summarized key ideas before ending each session.

Organization and logistics: The TBL sessions were held in the lecture hall of the University of Medicine and Pharmacy, Hue University. Each session involved one lecturer and one teaching assistant. The lecturer facilitated discussions, clarified issues, and guided problem-solving exercises, while the teaching assistant provided technical support during tests.

## Measurement instruments

To evaluate the effectiveness of the TBL method, the study employed four validated instruments

**Classroom Engagement Survey (CES).** Developed by Haidet P., the CES assessed student engagement in classroom activities and was administered both before and after the TBL course. The tool demonstrated Cronbach's alpha reliability of 0.80 [10].

**TBL-Student Assessment Instrument (TBL-SAI).** Designed by Heidi A. Mennenga (2012), the TBL-SAI measured individual contributions, learning preferences, and group satisfaction within TBL. This 33-item instrument, administered after the TBL course, demonstrated a high reliability with a Cronbach's alpha of 0.941 [11].

**Attitudes Towards TBL (ATL).** The ATL, created by Parmelee et al. (2009), evaluated satisfaction with group work, learning quality, team evaluations, clinical reasoning, and expertise development. This 19-item instrument was administered before and after the TBL course, achieving a Cronbach's alpha reliability of 0.947 [12].

**Perceived Collective Efficacy (PCE).** The PCE instrument developed by Salanova et al. (2003) evaluated perceptions of group efficacy before and after the TBL course. It consisted of four questions and demonstrated Cronbach's alpha reliability of 0.88 [4].

## Assessment process

The measurement scales were authorized for use by the original authors and were translated into Vietnamese following the guidelines of the World Health Organization [13]. Specifically, the scales were translated from English to Vietnamese by a translation team. The translation team consisted of four members with strong English proficiency and extensive experience working with the language, all of whom held IELTS certificates with scores above 6.0. The translation process followed a forward and backward translation approach. In the forward translation phase, two members translated the original English instruments into Vietnamese. The two translators then discussed and produced a preliminary Vietnamese version. In the backward translation phase, the research team provided the preliminary Vietnamese version to the other two members, who translated it back into English. The research team and the translation team then compared the back-translated version with the original English version, discussing and selecting the most appropriate wording to ensure the accuracy and cultural relevance of the Vietnamese translation.

The instruments' content validity was assessed using the Content Validity Index (CVI) with the participation of six experts in nursing and medical education. Each expert evaluated each item based on its relevance using a four-point scale (1 = not relevant, 4 = highly relevant). The Item-Level Content Validity Index (I-CVI) ranged from 0.83 to 1.00, meeting the acceptable standard (≥0.78). The Scale-Level Content Validity Index/Average (S-CVI/Ave) was 0.92, satisfying the content validity requirement (≥0.90). The Scale-Level Content Validity Index/Universal Agreement (S-CVI/UA) was 0.85, also within the acceptable range (≥0.80). These results indicate that the instrument has good content validity.

## Statistical analysis

The statistical analysis was carried out using SPSS software version 20.0 (IBM, Armonk, NY, USA). Frequencies and percentages were used to describe sample characteristics, and continuous variables are reported as means ± standard deviations (SD), along with maximum and minimum values. Missing data were excluded from the analysis. Differences in mean scores, including i-RAT and t-RAT scores as well as pre- and post-TBL course mean scores, were evaluated using paired t-tests. All probability values were two-sided, and p-values less than 0.05 were considered to indicate statistical significance.

Graphs were created using Microsoft Excel 2010 to visualize study variables. The statistical methods applied ensured accurate and reliable reporting of the data, consistent with best practices for analysis and interpretation.

## Results

### Characteristics of research subject

All 186 students completed the questionnaire, achieving a survey response rate of 100%. Results Table 1 shows that the average age of nursing students is 22.13 ± 0.656 years old. Among them, the proportion of female students (91.9%) is much higher than that of male students (8.1%). Detailed information regarding these indices is presented in Table 1.

### Average iRAT versus tRAT score of TBL sessions

Table 2 shows a clear difference in the average iRAT and tRAT scores of the three courses as well as all three courses. tRAT score is always higher than the iRAT score. This difference is statistically significant ($p < 0.05$).

Effectiveness of the TBL on Students' Attitudes Towards TBL, Classroom Engagement, and Perceived Collective EfficacyThe results of the study are summarized in Fig 1, which illustrates the impact of the TBL method on students' attitudes towards TBL, classroom engagement, and perceived collective efficacy. Significant improvements were observed in students' attitudes towards group learning after the TBL course, with increased scores in overall satisfaction with group work experience, group impact on learning quality, clinical reasoning ability, and professional development ($p < 0.001$). However, no significant change was noted in satisfaction with group member evaluation ($p = 0.061$). The mean CES score increased significantly from 29.54 ± 2.186 to 31.37 ± 2.002 after the course ($p < 0.001$). Additionally, students' perceptions of collective efficacy for group learning improved significantly after the TBL course ($p < 0.001$). This comprehensive visualization highlights the effectiveness of the TBL approach in enhancing attitudes, engagement, and collaborative learning among students.

**Table 1. Characteristics of research subjects.**

| Characteristics | | Quantity (n) | Proportion (%) |
|---|---|---|---|
| **Age** | Mean (SD) | **22.13 ± 0.656** | |
| **Gender** | Male | 15 | 8.1 |
| | Female | 171 | 91.9 |

**Table 2. Average iRAT versus tRAT score of TBL sessions.**

| Module | iRAT | tRAT | t | *P* |
|---|---|---|---|---|
| Module 1 | 7.99 | 9.76 | -15.111 | ***<0.001*** |
| Module 2 | 8.24 | 9.61 | -13.414 | ***<0.001*** |
| Module 3 | 8.28 | 9.66 | -15.573 | ***<0.001*** |
| Average 3 modules | 8.17 | 9.68 | -19.507 | ***<0.001*** |

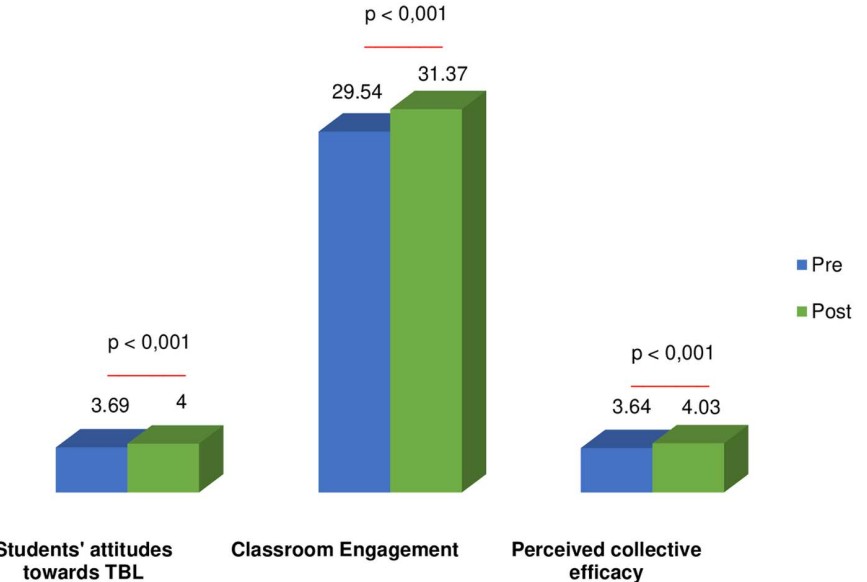

**Fig 1. Comparison of Students' Attitudes Towards TBL, Classroom Engagement and Perceived Collective Efficacy before and after the TBL course.**

## Mean Scores of TBL-SAI components and overall assessment

Table 3 shows the mean component scores and mean overall scores of the TBL-SAI. The average student assessment score on TBL (TBL-SAI) is $124.83 \pm 10.068$, of which, the average score on accountability is $31.19 \pm 2.975$, the average score for enjoyment of lectures or studying in class is $57.10 \pm 5.279$, and the average score for student satisfaction is $36.54 \pm 3.815$. These results show that students have positive experiences with the TBL learning method in all three areas.

## Discussion

The study was conducted on fourth-year nursing students at the University of Medicine and Pharmacy, Hue University. The participants had an average age of $22.12 \pm 0.66$ years and were predominantly female. t-RAT scores were significantly higher than i-RAT individual test scores. There were significant positive changes in attitudes toward group learning, perceptions of collective group effectiveness, and classroom participation after TBL compared to before TBL. In addition, TBL enhanced students' sense of accountability, interest in group learning, and overall satisfaction.

### The effectiveness of the TBL score

The tRAT scores in all three modules in the health care module for adults with Internal Medicine are significantly higher than their iRAT scores. Valeria Vanning's study of the impact of online TBL on the performance, attitudes, and

**Table 3. The mean TBL-SAI scores.**

| TBL-SAI | Mean±SD | Min-Max |
|---|---|---|
| Accountability Subscale | 31.19±2.975 | 23-40 |
| Preference for Lecture or TBL Subscale | 57.10±5.279 | 46-76 |
| Student Satisfaction Subscale | 36.54±3.815 | 28-45 |
| **Overall (TBL-SAI)** | **124.83±10.067** | **103-161** |

accountability of undergraduate nursing students during the COVID-19 pandemic in 2022 showed significant improvement between tRAT and iRAT performance in all online TBL sessions (p < 0.001) [14]. Results of testing students' readiness with TBL in a meta-analysis study by Phan Nguyen Ngoc and colleagues (2020) with 11 studies and 1575 participants who are students in the health care professions showed significantly higher tTRAT scores than iRAT scores. In particular, the tRAT/iRAT ratio in the nursing student group was much better than in groups composed of students from different majors [15]. The average tRAT score significantly improved compared to the i-RAT score, highlighting better performance in group assignments than individual work. This progression from iRAT to tRAT reflects students' initial success in the TBL process, even before receiving instructor guidance. The results underscore the positive impact of TBL on learning outcomes, driven by effective group communication and collaboration. Previous research has also shown that learners perform better on the tRAT if they complete the iRAT first because they will need to consider the questions themselves first and are more likely to make an effort to participate in the discussion to share their views as well as listen to other people's opinions to reach a consensus on test answers [16]. The TBL course design promotes a collaborative learning environment where students engage in discussions to solve t-RAT questions and address practical scenarios relevant to patient care. This approach allows gaps in individual knowledge to be resolved through group consensus, enhancing understanding. The process emphasizes teamwork and interaction, key skills for students' future professional practice. During cooperative learning, students discussed how to apply knowledge strategically to improve their group's scores. Task interdependence enhanced team members' ability to perform tasks and integrate their ideas during group discussions [17].

## Students' attitudes towards TBL

Assessing students' attitudes when applying new teaching methods is essential to help educators design appropriate teaching models. We used ATL instruments to explore students' attitude toward TBL. The average attitude score of nursing students after taking TBL increased significantly compared to before the TBL course. The overall average score of satisfaction with teamwork experience increased from $3.69 \pm 0.460$ before learning TBL to $4.00 \pm 0.468$ after learning TBL (p < 0.001). This result shows that after experiencing the TBL course, students have a more positive attitude toward the group work experience than before applying the traditional method with lectures. Research by Valeria Vannini (2019) also showed positive results on overall satisfaction with group work experience, with a score from $3.78 \pm 0.6$ before to $4.19 \pm 0.8$ after TBL intervention [14]. It is possible that participating in student groups in each module has helped them gradually become more proficient in working in groups and become more contributing members in the TBL learning process. This may have also helped students find more value in the group experience. Significant positive shifts in students' attitudes toward team learning were observed after the TBL course in our study. Students perceived TBL as more effective compared to previous methods, citing improvements in clinical reasoning, self-awareness, collaborative leadership, and respect for diverse perspectives. TBL's structured approach, emphasizing group work and collective testing, fostered cooperation and cohesion. By collaborating to solve simulated clinical scenarios, students enhanced their knowledge, communication, and decision-making skills. However, attitudes toward peer assessment did not change significantly from pre- to post-TBL. This area received the lowest satisfaction scores among five evaluated areas. Similarly, Vannini et al. (2022) reported a non-significant decrease in satisfaction with peer assessment, from $3.59 \pm 0.6$ to $3.38 \pm 0.9$ [14]. Peer assessment, where group members evaluate each other's contributions, is a key component of TBL for improving group performance and developing individual and teamwork skills necessary for healthcare practice. It also enhances students' feedback-giving abilities and provides instructors with data for fair grading by assessing each member's contribution to the group. This skill is crucial in healthcare, where professionals are frequently evaluated by patients, peers, and colleagues [18]. Studies suggest that many students dislike peer assessment [12,19,20], likely due to discomfort or lack of familiarity. Govindarajan and Rajaragupathy's research on online TBL in Biochemistry also found that receiving constructive peer feedback had the lowest satisfaction scores [21]. Effective implementation requires thorough training to help students

understand the feedback process [22]. Yet, discomfort in assessing peers suggests that peer assessment should be introduced and taught prior to implementing TBL.

### The effectiveness of the TBL on students' class participation level

Classroom participation significantly improved after the TBL course (p < 0.001), consistent with Isabel McMullen et al.'s study on UK psychotherapists [23], showing that group interaction and collaboration foster greater student engagement.

These findings align with reports confirming TBL's role in fostering collaborative learning environments [24]. By emphasizing team readiness and active group activities, TBL promotes student engagement, encouraging more active participation in exercises and activities.

### Students' perceptions of collective effectiveness when studying in groups

Our study revealed a significant improvement in students' perception of collective effectiveness after the TBL course, consistent with Wong et al. (2017) [25]. This highlights students' increased appreciation for group learning opportunities. TBL enhances collective efficacy by fostering task interdependence, encouraging discussion, integrative thinking, and reliance on colleagues' input. These skills help nursing students develop teamwork abilities essential for providing optimal patient care. In healthcare, collective efficacy is linked to reduced missed care, improved patient outcomes [26], and better nursing performance [27]. Ganotice Jr. et al. (2022) also emphasized its critical role in successful team collaboration [28].

### Student assessments of TBL

The overall mean score and subscale scores for student assessments of TBL on the TBL-SAI scale were above neutral, indicating that students felt a sense of responsibility, enjoyment, satisfaction, and preference for TBL-based courses. Similar findings were reported by Tan et al. (2021) and Burton (2021), highlighting TBL's positive influence on learning experiences [29,30]. The accountability score in this study (31.19 ± 2.975) reflected students' strong preparation and contributions to their teams. Students preferred TBL over traditional lectures due to its emphasis on teamwork, problem-solving, and instructor feedback. While Ibrahim (2020) found mixed results, our study shows TBL improves retention and test performance by enhancing active participation and collaborative learning [31]. Students reported high satisfaction with TBL (36.54 ± 3.815), attributed to its interactive, small-group learning environment. Peer collaboration and skill development contributed to positive learning experiences. Mangold (2007) emphasized aligning teaching strategies with student preferences for optimal outcomes [32].

## Limitations of the study

This study involved the same group of participants who had previously experienced traditional learning methods in other subjects. As such, confounding factors, such as the appeal of the course content and the instructor's presentation skills, may have influenced the outcomes. Additionally, since all students participated in group learning within this module, the study could not identify differences in scores between groups using TBL and those not engaged in group learning methods.

## Conclusions

In this study, the TBL course design was effectively implemented in the Nursing subject of caring for adults with Internal Medicine diseases. TBL improves student learning outcomes, student accountability, interest, and satisfaction. Students' attitudes toward group learning improved after the TBL course. Student participation is high in the TBL classroom. Students' awareness of collective effectiveness also improved significantly. TBL enhances student-teacher interaction, is very useful, and is a viable active learning strategy for large student-to-faculty ratios.

## Supporting information

**S1 Dataset.  Team-Based Learning Effectiveness dataset.**
(SAV)

## Acknowledgments

We would like to acknowledge the team members for their invaluable contribution from conception to the final approval for submission to publication. We sincerely thank the University of Medicine and Pharmacy, Hue University, and the Faculty of Nursing lecturers for their support and encouragement in completing this study.

## Author contributions

**Conceptualization:** Lan Duong Thi Ngoc.

**Data curation:** Lan Duong Thi Ngoc, Anh Nguyen Ngoc Quynh, Thao Hoang Thi Phuong.

**Formal analysis:** Lan Duong Thi Ngoc, Anh Nguyen Ngoc Quynh, Thao Hoang Thi Phuong.

**Funding acquisition:** Lan Duong Thi Ngoc, Anh Nguyen Ngoc Quynh, Thao Hoang Thi Phuong.

**Investigation:** Lan Duong Thi Ngoc, Anh Nguyen Ngoc Quynh, Thao Hoang Thi Phuong.

**Methodology:** Lan Duong Thi Ngoc, Anh Nguyen Ngoc Quynh, Thao Hoang Thi Phuong.

**Project administration:** Lan Duong Thi Ngoc, Anh Nguyen Ngoc Quynh, Thao Hoang Thi Phuong.

**Resources:** Lan Duong Thi Ngoc, Anh Nguyen Ngoc Quynh, Thao Hoang Thi Phuong.

**Software:** Lan Duong Thi Ngoc, Anh Nguyen Ngoc Quynh, Thao Hoang Thi Phuong.

**Supervision:** Lan Duong Thi Ngoc.

**Validation:** Lan Duong Thi Ngoc, Anh Nguyen Ngoc Quynh, Thao Hoang Thi Phuong.

**Visualization:** Lan Duong Thi Ngoc, Anh Nguyen Ngoc Quynh, Thao Hoang Thi Phuong.

**Writing – original draft:** Lan Duong Thi Ngoc, Anh Nguyen Ngoc Quynh, Thao Hoang Thi Phuong.

**Writing – review & editing:** Lan Duong Thi Ngoc, Anh Nguyen Ngoc Quynh, Thao Hoang Thi Phuong.

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
