## [Decision Letter · Decision Letter 0]

10 Mar 2025

PONE-D-25-05778A Pre – Post quasi-experimental study of Team-Based Learning Effectiveness for Vietnamese Nursing StudentsPLOS ONE

Dear Dr. Nguyen Ngoc,

Thank you for submitting your manuscript to PLOS ONE. After careful consideration, we feel that it has merit but does not fully meet PLOS ONE’s publication criteria as it currently stands. Therefore, we invite you to submit a revised version of the manuscript that addresses the points raised during the review process.

We look forward to receiving your revised manuscript.

Kind regards,

Mukhtiar Baig, Ph.D.

Academic Editor

PLOS ONE

Journal Requirements:

2. Please remove all personal information, ensure that the data shared are in accordance with participant consent, and re-upload a fully anonymized data set.

Reviewers' comments:

Reviewer's Responses to Questions

**Comments to the Author**

1. Is the manuscript technically sound, and do the data support the conclusions?

Reviewer #1: Partly

Reviewer #2: Yes

2. Has the statistical analysis been performed appropriately and rigorously? 

Reviewer #1: I Don't Know

Reviewer #2: Yes

3. Have the authors made all data underlying the findings in their manuscript fully available?

Reviewer #1: Yes

Reviewer #2: Yes

4. Is the manuscript presented in an intelligible fashion and written in standard English?

Reviewer #1: Yes

Reviewer #2: Yes

5. Review Comments to the Author

Reviewer #1: Dear authors

your topic is good, but it is recommended:

1- improve the introduction with newer study

2- firs paragraph of introduction is too long.

3- highlight importance of using TBL is good, but you have not integration in introduction.

4- the end of introduction is better you note your goal of study.

Method:

1- note how many students were excluded and why? How many was Research population?

2- describe your course: how many section or hours? About course class how many classes and how many students were in each class?

Be success

Reviewer #2: In the last paragraph of the introduction, we should see the study objective. Please clearly mention the objectives in the text.

The process for translation and validation of tools are unclear.

Did the normality condition was checked for all tools? What was the statistical parameters for checking the normality?

6. PLOS authors have the option to publish the peer review history of their article (what does this mean? ). If published, this will include your full peer review and any attached files.

**Do you want your identity to be public for this peer review?** For information about this choice, including consent withdrawal, please see our Privacy Policy .

Reviewer #1: No

Reviewer #2: No

---

## [Author Response · Author response to Decision Letter 1]

28 Mar 2025

1. Journal Requirements: We have edited as requested

2. Reviewer 1:

- Introduction: We have updated the latest research; We have shortened and ensured the balance of the length of the paragraphs; We have removed the section integrating the use of TBL; We have added the research objectives at the end of introduction.

- Method: All 186 students participated in the study, with no students excluded; We have added the requested course description into the manuscript.

3. Review 2

- We have added the research objectives at the end of introduction

- The translation followed the World Health Organization (WHO) guidelines, including forward translation, backward translation, and expert review to ensure linguistic and conceptual equivalence. Additionally, the content validity of the instrument has been incorporated into the manuscript

- Yes, We checked the normality condition, and the variables did not follow a normal distribution. We used the Q-Q plot method to assess the normality of the variables in this study.

---

## [Editor Report · Decision Letter 1]

13 Apr 2025

A pre – post quasi-experimental study of team-based learning effectiveness for Vietnamese nursing students

PONE-D-25-05778R1

Dear Dr. Ngoc,

We’re pleased to inform you that your manuscript has been judged scientifically suitable for publication and will be formally accepted for publication once it meets all outstanding technical requirements.

Kind regards,

Mukhtiar Baig, Ph.D.

Academic Editor

PLOS ONE

---

## [Editor Report · Acceptance letter]

PONE-D-25-05778R1

PLOS ONE

Dear Dr. Nguyen Ngoc Quynh,

I'm pleased to inform you that your manuscript has been deemed suitable for publication in PLOS ONE. Congratulations! Your manuscript is now being handed over to our production team.

Kind regards,

on behalf of

Professor Mukhtiar Baig

Academic Editor

PLOS ONE